# Can Surgical Resection of Metastatic Lesions Be Beneficial to Pancreatic Ductal Adenocarcinoma Patients with Isolated Lung Metastasis?

**DOI:** 10.3390/cancers14092067

**Published:** 2022-04-20

**Authors:** Won-Gun Yun, Wooil Kwon, Youngmin Han, Hee Ju Sohn, Hyeong Seok Kim, Mirang Lee, Hongbeom Kim, Alexander S. Thomas, Michael D. Kluger, Jin-Young Jang

**Affiliations:** 1Department of Surgery and Cancer Research Institute, College of Medicine, Seoul National University, 101 Daehak-ro, Chongno-gu, Seoul 03080, Korea; wkeonyun5@naver.com (W.-G.Y.); willdoc@snu.ac.kr (W.K.); views@snu.ac.kr (Y.H.); generalheeju@gmail.com (H.J.S.); ucla0701@snu.ac.kr (H.S.K.); rang5026@snu.ac.kr (M.L.); sugeonkhb@snu.ac.kr (H.K.); 2Division of Gastrointestinal and Endocrine Surgery, Department of Surgery, Columbia University Vagelos College of Physicians and Surgeons, New York, NY 10032, USA; at3215@cumc.columbia.edu (A.S.T.); mk2462@cumc.columbia.edu (M.D.K.)

**Keywords:** pancreatic ductal adenocarcinoma, metastasis/lung, metastatectomy, chemotherapy, survival

## Abstract

**Simple Summary:**

With the development of chemotherapy, studies have been conducted on the possibility of conversion surgery in metastatic pancreatic ductal adenocarcinoma (PDAC) patients. In addition, studies with large-scale public data have reported that patients with isolated lung metastasis have a good prognosis and can benefit from survival through surgical treatment. Our study aims to evaluate the effect of metastatectomy and prognostic factors in PDAC patients with isolated lung metastasis by analyzing the data from 1342 patients in our institution. We showed that PDAC patients with isolated lung metastasis who underwent metastatectomy seemed to have better survival when compared with patients who underwent only chemotherapy or supportive care. In addition, we performed the analysis using the National Cancer Database for external validation purposes and found consistent results compared to our analysis. Our findings suggest that PDAC patients with isolated lung metastasis should be considered for multimodal therapy with chemotherapy and surgical treatment.

**Abstract:**

In the era of effective chemotherapy on pancreatic ductal adenocarcinoma (PDAC) with distant metastasis, data on the effects of metastatectomy are lacking. So, we investigated the effect of metastatectomy on survival after metastasis in PDAC patients with isolated lung metastasis. This retrospective study analyzed 1342 patients who were histologically diagnosed with PDAC with distant metastasis from January 2007 to December 2018, of which 83 patients had isolated pulmonary metastasis. Additionally, 4263 patients were extracted from the National Cancer Database (NCDB) and analyzed. Log-rank test and Kaplan−Meier survival analysis were used to analyze survival after metastasis. The five-year survival rate was significantly higher in patients who underwent pulmonary metastatectomy than in those who received only chemotherapy or supportive treatment (60.6% vs. 6.2% vs. 0.0%, *p* < 0.001). A similar trend was observed in the NCDB (two-year survival rate, 27.4% vs. 15.8% vs. 4.7%, *p* < 0.001). In the multivariate analysis, lung lesion multiplicity (hazard ratio (HR) = 2.004, *p* = 0.017), metastatectomy (HR = 0.278, *p* = 0.036), chemotherapy (HR = 0.434, *p* = 0.024), and chemotherapy cycles (HR = 0.300, *p* < 0.001) had significant effects on survival. Metastatectomy with primary pancreatic lesions is recommended with effective chemotherapy in PDAC patients with isolated lung metastasis.

## 1. Introduction

Pancreatic cancer, a major global health problem, is the fourteenth most common cancer in men, thirteenth most common cancer in women, and seventh leading cause of cancer death in both sexes worldwide because of its poor prognosis, according to the Global Cancer Statistics 2020 [1]. Surgery is important, because it remains the only treatment that offers a curative potential, although <20% of patients are diagnosed with resectable disease [2]. Despite several early diagnostic techniques, approximately 50% of patients present with distant metastasis at the time of diagnosis. According to the 2021 National Comprehensive Cancer Network clinical practice guideline [3], systemic treatments, such as chemotherapy or best supportive care, are the only recommended treatments for pancreatic ductal adenocarcinoma (PDAC) with distant metastasis, regardless of the metastatic pattern or specific site(s), because the primary goals of treatment are palliation and lengthened survival. Recent studies have suggested that surgical treatment may also be considered for PDAC with distant metastasis in clinically specific cases [4,5,6,7].

Since 2010, there have been significant advances in chemotherapy regimens for PDAC. Fluorouracil (5-FU) plus leucovorin, irinotecan, and oxaliplatin (FOLFIRINOX), or gemcitabine plus albumin-bound paclitaxel (GEM/Ab), are the preferred therapies in metastatic PDAC patients with a good performance status and have provided a challenge in patient survival expectations [8,9,10,11,12,13]. With the development of chemotherapy, studies have been conducted on the possibility of conversion surgery in selected patients who respond extremely well to chemotherapy. According to Frigerio et al. [4], PDAC patients with liver metastasis who have a good response to neoadjuvant chemotherapy may be cautiously considered for surgery, with a potential benefit in survival compared with palliative chemotherapy alone. According to Byun et al. [5], patients treated with surgery after at least four cycles of FOLFIRINOX had a significantly longer median survival than those with synchronous distant metastasis in the non-surgical group (32 months vs. 14 months).

A few retrospective studies on the impacts of surgery on survival in PDAC patients with distant metastasis suggest that those with synchronous or metachronous single-organ distant metastasis may have a different prognosis depending on the location of metastasis, and that surgery can be beneficial in patients with isolated lung metastasis [6,7,14,15].

Although large-scale evidence is lacking, the few studies on the topic suggest that PDAC patients with isolated lung metastasis have an exceptionally good prognosis and that they may benefit from surgery [6,7,14,15]. We aimed to investigate the effect of metastatectomy on survival after metastasis in PDAC patients with isolated lung metastasis.

## 2. Materials and Methods

### 2.1. Study Population

Between January 2007 and December 2018, 2023 patients at the Seoul National University Hospital were diagnosed with primary pancreatic cancer with distant metastasis. There were 1285 patients with synchronous metastasis, and 738 had metachronous metastasis. Synchronous metastasis was defined as metastasis confirmed at the time of diagnosis of primary PDAC, and metachronous metastasis was defined as metastasis confirmed during the follow-up period after treatment, including surgery of primary PDAC. Among the patients with synchronous metastasis, those who were not histologically confirmed or diagnosed with lesions other than PDAC (*n* = 245), who underwent surgery with no curative intent (*n* = 132), who were lost to follow-up or had treatment at an outside hospital (*n* = 63), and those who died within 3 months after initial surgery (*n* = 2) were excluded. Among the patients with metachronous metastasis, those who underwent surgery with no curative intent (*n* = 82), who were histologically diagnosed with lesions other than PDAC (*n* = 67), who were lost to follow-up or had treatment at an outside hospital (*n* = 45), who died within 3 months after initial surgery (*n* = 27), and who were diagnosed with other cancers simultaneously (*n* = 18) were excluded.

We retrospectively reviewed the prospectively collected electronic medical records of 1342 qualifying patients. Among them, 83 had isolated pulmonary metastasis. We classified these 83 patients according to the metastatic pattern and treatment option (Figure 1).

### 2.2. Data Collection

The clinical and radiological data were collected from the patient medical records. We collected factors associated with demographics, primary PDAC lesions, metastatic lesions, and treatment options that were thought to influence the survival outcome of patients. These included age, sex, primary tumor location, primary tumor size at diagnosis, suspicious regional lymph node metastasis at initial imaging, date of diagnosis, metastatic sites and patterns, multiplicity of metastatic lesions, carcinoembryonic antigen level and carbohydrate antigen (CA) 19-9 level at the time metastasis was confirmed, date of confirmed metastasis, chemotherapy regimen, chemotherapy cycle, response to chemotherapy, radiotherapy for the metastatic lesion, and date of death or last follow-up [16,17,18,19,20,21,22,23,24,25,26,27].

The survival period was based on the date of death or the last visit to the hospital from the time metastatic lung lesions were identified. Metastasis was diagnosed through biopsy or serial imaging exams including computed tomography, magnetic resonance imaging, and positron emission tomography with CA 19-9. Chemotherapy response assessment was performed based on computed tomography according to the revised response evaluation criteria in solid tumors, version 1.1, and each case was classified as complete response (CR), partial response (PR), stable disease (SD), or progressive disease (PD) [28].

### 2.3. National Cancer Data Base (NCDB) Mining Strategy for Validation

The clinical, surgical, and survival information of PDAC patients with distant metastasis was extracted from the NCDB (2010–2016). We found 4691 histologically diagnosed PDAC patients with isolated pulmonary metastasis. We excluded patients who had insufficient information about the resection of metastatic lesions, chemotherapy, and supportive care (*n* = 406), and those who died within 3 months after surgery (*n* = 22). We categorized 4263 patients according to the treatment options (Figure 2).

### 2.4. Statistical Analysis

All of the statistical analyses were performed using IBM SPSS software for Windows (version 25.0; IBM Corp., Armonk, NY, USA) and R software, version 4.1.2 (R Foundation for Statistical Computing). Categorical variables were analyzed using the chi-square test and Fisher’s exact test.

A survival analysis was performed according to the clinical characteristics and treatment options using the Kaplan–Meier analysis and log-rank test. The log-rank test and Cox proportional hazard model were used to explore the factors that significantly affected survival; *p*-values < 0.050 were considered statistically significant.

## 3. Results

### 3.1. Demographics

The demographics of the 83 PDAC patients with isolated pulmonary metastasis included in this study are shown in Table 1. Among them, 39 (47.0%) were males, with a mean age of 64.5 years. Depending on the tumor location, 42 (50.6%) were found to have tumors on the pancreatic head and 41 (49.4%) on the body/tail. The mean size of PDAC at initial diagnosis was 28.1 mm.

Regarding surgical treatment, 15 (18.1%) patients underwent metastatectomy within 6 months after metastasis. Of them, two had synchronous metastasis and thirteen had metachronous metastasis. One of the patients with synchronous metastasis underwent video-assisted thoracic surgery (VATS) with curative intent for diagnostic confirmation of lung lesions, along with surgery for primary pancreatic cancer simultaneously. Another patient received chemotherapy without pancreatectomy after confirming that the lung lesions were metastases of the pancreas through VATS. Thirteen patients with metachronous metastasis underwent VATS for lung lesions that were discovered during the follow-up period. Subsequently, eleven patients received chemotherapy; two were followed without receiving additional systemic treatment.

Regarding chemotherapy after metastasis, 14 (16.9%) patients did not receive chemotherapy, 10 (12.0%) received gemcitabine monotherapy, 20 (24.1%) received gemcitabine combined with agents other than albumin-bound paclitaxel, 13 (15.7%) received GEM/Ab, 24 (28.9%) received FOLFIRINOX, and 2 (2.4%) received other chemotherapy regimens (TS-1 and 5-FU based). The median number of chemotherapy cycles was six. Depending on the response to chemotherapy after at least two cycles, the number of cases of PR, SD, and PD was 11 (16.2%), 39 (57.4%), and 13 (19.1%), respectively. Only one patient received radiotherapy for metastasis.

The demographics were compared between patients with metachronous and synchronous metastasis, with single, oligometastases, and multiple metastases (Appendix A). Oligometastases were defined as having two to five lesions, based on previous reports on the local treatment of pulmonary metastatic lesions [29,30,31]. The proportion of patients with elevated CA 19-9 levels was significantly higher among patients with synchronous metastasis (metachronous 24.4% vs. synchronous 92.3%, *p* = 0.001). There were no significant differences in other demographic factors, except CA 19-9 elevation, between the metachronous and synchronous metastasis groups, and all demographic factors among groups according to the multiplicity of metastatic lesions.

### 3.2. Survival Analysis

The median survival of 83 patients after pulmonary metastasis was 19 months, and the two-year and five-year survival rates were 35.1% and 10.7%, respectively. The median survival was 7 months in 1259 patients with distant metastasis of sites other than the lung, the two-year survival rate was 8.4%, and the five-year survival rate was 1.2%. PDAC patients with isolated lung metastasis showed a significantly better prognosis than those with distant metastasis at other sites (*p* < 0.001) (Figure 3a).

In the survival analysis based on the temporal metastatic pattern, the median survival was 20 and 14.5 months in patients with metachronous and synchronous metastasis, respectively (*p* = 0.014) (Figure 3b). According to the multiplicity of metastatic lung lesions, the median survival was 22 months in patients with a single metastasis, 19 months in patients with oligometastases (2~5 lesions), and 14 months in patients with multiple (>5 lesions) metastases (*p* = 0.019) (Figure 3c). As such, prognosis seemed directly related to the number of metastatic lesions; it was not statistically significant in the comparison between the two groups, except for single vs. multiple (single vs. oligometastases, *p* = 0.064, oligometastases vs. multiple, *p* = 0.284, single vs. multiple, *p* = 0.007).

When survival was analyzed by treatment type, there were statistically significant differences in the five-year survival rate among patients who underwent metastatectomy with or without chemotherapy and those who received only chemotherapy or supportive care (60.6% vs. 6.2% vs. 0.0%, *p* < 0.001) (Figure 4a). Among the patients who did not undergo surgery, the median survival was 20 months for FOLFIRINOX or GEM/Ab, 17 months for gemcitabine combined with agents other than albumin-bound paclitaxel, 15 months for gemcitabine monotherapy, and 9 months for supportive care only (*p* < 0.001) (Figure 4b). In the comparison between the two groups, FOLFIRINOX or GEM/Ab and gemcitabine combined with agents other than albumin-bound paclitaxel had a better prognosis than gemcitabine monotherapy in terms of the median survival, and the two-year and five-year survival rates, respectively; however, the comparison between gemcitabine combined with agents other than albumin-bound paclitaxel and gemcitabine monotherapy was not statistically significant (FOLFIRINOX or GEM/Ab vs. gemcitabine monotherapy, *p* = 0.015, gemcitabine combined with agents other than albumin-bound paclitaxel vs. gemcitabine monotherapy, *p* = 0.177).

### 3.3. External Validation with NCDB

Median survival in the cohort of 4691 patients from the NCDB was 14 months for patients treated with metastatectomy with or without chemotherapy, 10 months for those treated with chemotherapy only, and 2 months for supportive care only (*p* < 0.001) (Figure 5a). Among the patients who did not undergo metastatectomy, the median survival was 12 months for patients treated with multi-agent regimens, 7 months for single-agent regimens, and 2 months for supportive care (*p* < 0.001) (Figure 5b).

### 3.4. Prognostic Factors

The univariate and multivariate analyses for survival after isolated pulmonary metastasis are shown in Table 2. In the univariate analysis, demographic factors and primary pancreatic lesion characteristics were not significant factors for survival outcomes. Metastatic lesion multiplicities (hazard ratio (HR) = 1.811, *p* = 0.025), metastatectomy (HR = 0.183, *p* = 0.004), or chemotherapy (HR = 0.265, *p* < 0.001) for metastasis, and number of chemotherapy cycles (HR = 0.287, *p* < 0.001) were statistically significant factors.

In the multivariate Cox proportional hazards model analysis, metastatic lesion multiplicities (HR = 2.004, *p* = 0.017), metastatectomy (HR = 0.278, *p* = 0.036), or chemotherapy (HR = 0.434, *p* = 0.024) for metastasis, and number of chemotherapy cycles (HR = 0.300, *p* < 0.001) were significant prognostic factors for survival after metastasis.

## 4. Discussion

Detecting pancreatic cancer at the resectable point is challenging. Although surgery introduces potential complications, it remains the only curative treatment for pancreatic cancer, making it important to identify patients who can benefit from it. Local treatments, such as surgery, are currently not recommended by guidelines in the case of distant metastasis in most cancers, because patients are thought to not benefit from surgery. As the systemic effects of chemotherapy have recently improved, pulmonary metastatectomy has been reported to improve survival in certain cancers such as colorectal cancer and hepatocellular carcinoma [32,33,34,35,36,37,38]. The few reports suggesting the benefit of metastatectomy for PDAC patients with isolated lung metastasis have inherent limitations. According to Liu et al. [6], in PDAC patients with synchronous lung metastasis, resection of the primary pancreatic tumor achieved relatively longer survival following resection with or without metastatectomy compared with no primary pancreas resection (14.0 months vs. 6.0 months; *p* < 0.001). According to Kim et al. [7], in patients with metachronous lung metastasis from PDAC, survival after metastasis was significantly longer for patients who underwent metastatectomy (36.5 months vs. 9.5 months; *p* = 0.010). Herein, 83 patients diagnosed with PDAC with isolated lung metastasis were analyzed to evaluate the effect of various treatments on survival. We observed that pulmonary metastatectomy and multi-agent chemotherapy regimens were beneficial, and that metachronous metastatic patterns were associated with a good prognosis. Survival was also directly related to the number of metastases. We conducted an external validation using the NCDB to overcome the limitations of sample size at a single center. In the survival analysis using the NCDB, we observed that pulmonary metastatectomy and multiple chemotherapy agents were associated with improved survival. These results were consistent with those of our primary cohort.

Based on the temporal metastatic pattern, we found that PDAC patients with metachronous lung metastasis had better survival rates than those with synchronous lung metastasis. Although direct comparisons have not been made between PDAC patients with synchronous and metachronous lung metastasis, the survival period of patients with metachronous metastasis who underwent metastatectomy appears to be much longer than that of patients with synchronous metastasis, even in a paper based on the SEER database [6]. We suspect that the tumor burden at the time of diagnosis of metastasis was related to these outcomes in to previous studies. According to Wei, T. et al. [39], circulating tumor DNA (ctDNA) mutant allele frequency (MAF) is correlated with tumor burden, and found PDAC patients with ctDNA MAF more than 1.5% had a poor prognosis. In addition, several studies have shown that the amount of ctDNA is associated with the prognosis in PDAC patients [40,41,42]. In our institution’s data, 33.3% (19/57) had multiple lung metastases in patients with metachronous lung metastasis, whereas 46.2% (12/26) had multiple lung metastases in patients with synchronous lung metastasis at the time metastasis was diagnosed. In addition, patients with synchronous metastasis also have primary pancreatic lesions at the time metastasis is detected, so the tumor burden will be higher. To elucidate this, it is thought that studies on the biological differences of metastatic lesions according to the temporal metastatic pattern will be needed in the future, considering the factors related to chemotherapy.

In our study, patients who received chemotherapy had significantly better survival rates than those who did not. Patients who received multi-agent chemotherapy regimens, perhaps introducing some bias based on their underlying performance status, such as FOLFIRINOX or GEM/Ab, had a better prognosis than patients treated with gemcitabine monotherapy. Among the patients who received only chemotherapy, the objective response rate was 26.3% in the FOLFIRINOX group, 22.2% in the GEM/Ab group, and 16.6% in the gemcitabine monotherapy group. Prior studies of metastatic pancreatic cancer have reported a response rate of 31.6% for FOLFIRINOX, 23% for GEM/Ab, and <10% for gemcitabine monotherapy [10,11]. These results suggest that, although PDAC patients with isolated lung metastasis may benefit from chemotherapy, the exceptionally good prognosis is not attributable to only the chemotherapy effect.

If we look at the recurrence patterns after pancreatectomy in PDAC patients with metachronous metastasis, systemic recurrence was seen in 73.5% of patients, among which the liver was the most common site, but the lung also had a high proportion (Appendix A). In addition, the median recurrence-free survival for who it was isolated and recurred at lung was 18 months, which was significantly longer than those patients for who it recurred elsewhere (Appendix A). Considering the survival benefits from metastatectomy and the findings suggesting an indolent nature, it can be assumed that PDAC patients with isolated pulmonary metastasis are biologically different from other patients. According to Aramcki et al. [43], PDAC patients with isolated pulmonary metastasis are a distinct clinical and genetic subgroup, and tumors with lung metastasis display a significantly lower protein kinase D1 expression, which induces an increase in the secretion of small extracellular vesicles from cancer cells in PDAC compared with patients with liver metastasis. According to Nimmakayala et al. [44], distinct cancer stem cell (CSC) subtypes are strongly associated with organ-specific colonization. Liver metastasis showed drug-resistant CSC and an epithelial-to-mesenchymal transition-like phenotype; lung metastasis displayed an aldehyde dehydrogenase-positive/CD133-positive subpopulation CSC and mesenchymal-to-epithelial transition-like phenotype. In our study, the clinical attributes of the primary PDAC lesions were not significantly associated with survival, while the treatment modality after metastasis occurrence and multiplicity of metastatic lesions were. A previous study on a prognostic nomogram for PDAC patients with lung metastasis through the Surveillance, Epidemiology, and End Results database showed that surgery or chemotherapy had the most significant impact on survival, and that the location of primary pancreatic lesion (head or body/tail) within the pancreas was the only significant prognostic factor related to the primary pancreatic lesion [45]. This implies that the clinically verifiable phenotype is not sufficient to reflect the genotype associated with specific organ metastasis, and a system to classify PDAC with a genotype and biomarker needs to be developed. For example, it is known that the Kras mutation is associated with the growth of not only primary pancreatic lesions, but also metastatic lesions, as found in several studies [46,47]. Therefore, a deeper understanding of lung metastasis will be possible by evaluating the difference in gene expression between primary and metastatic lesions.

In our study, based on the multiplicity of metastatic lung lesions, the number of metastatic lesions was associated with prognosis. Patients with single metastasis had a significantly better prognosis than those with multiple metastases, and even showed a trend toward better survival than those with oligometastases. These results suggest that the lower the number of metastatic lesions, the more aggressive the treatment should be. Based on this study, complete resection of metastatic lesions through VATS can be recommended, especially in patients with single or oligometastases with a distribution suitable for surgery. In addition to VATS, stereotactic ablative radiotherapy (SABR) may be considered as an alternative for patients in whom surgically resecting metastatic lesions is challenging due to the distribution of lesions or the performance status [48,49]. It has been reported that SABR for oligometastatic lung lesions from various primary cancers is safe and is associated with a high rate of local control [50]. Herein, only one patient received radiotherapy for metastatic lung lesions. The number of patients was too small for a meaningful analysis, and this patient attempted SABR as an additional treatment option after the metastatic lesions were not controlled by chemotherapy and the performance status deteriorated. Further large-scale studies are needed for clinical applications of these local treatments on pulmonary metastatic lesions.

Our study has limitations. First, it was a retrospective single-center study, and the sample size was relatively limited. Second, there was no evaluation of biological factors. To overcome this, future studies should ideally integrate surgeons, pathologists, geneticists, and basic scientists. Third, this study did not evaluate the ideal timing of surgery for metastatic lung lesions. Although all patients in this study underwent metastatectomy within 6 months of metastasis, this is insufficient to suggest an optimal timing of metastatectomy. It should be carefully determined through a multidisciplinary approach.

## 5. Conclusions

This study shows the effect of metastatectomy on survival rates for PDAC patients with synchronous or metachronous lung metastasis. The findings suggest that PDAC patients with isolated lung metastasis should be considered for multimodal therapy with chemotherapy and surgical resection of both the primary pancreas and metastatic lung lesions. Future clinical studies are required to identify the optimal treatment method, treatment period, and chemotherapy regimens for PDAC patients with isolated lung metastasis, and studies related to tumor biology and SABR should be considered.

## Figures and Tables

**Figure 1 cancers-14-02067-f001:**
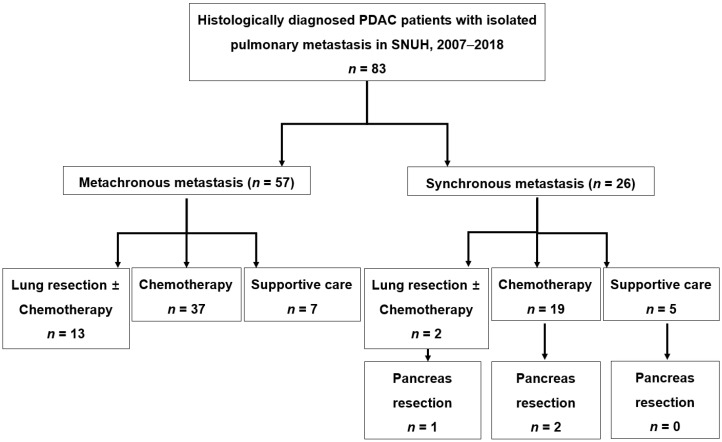
Flow diagram of the study with 83 patients who were histologically diagnosed as having pancreatic ductal adenocarcinoma with isolated pulmonary metastasis between January 2007 and December 2018.

**Figure 2 cancers-14-02067-f002:**
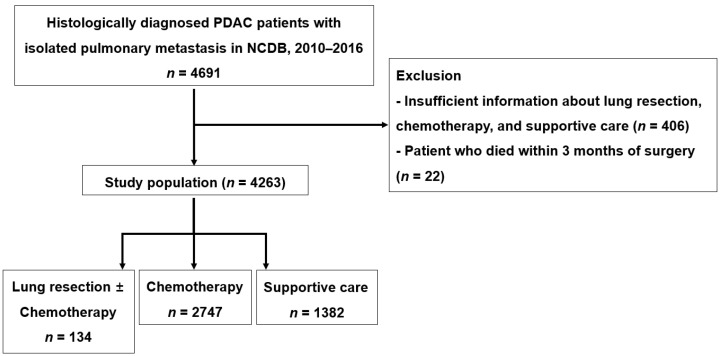
National Cancer Database mining strategy.

**Figure 3 cancers-14-02067-f003:**
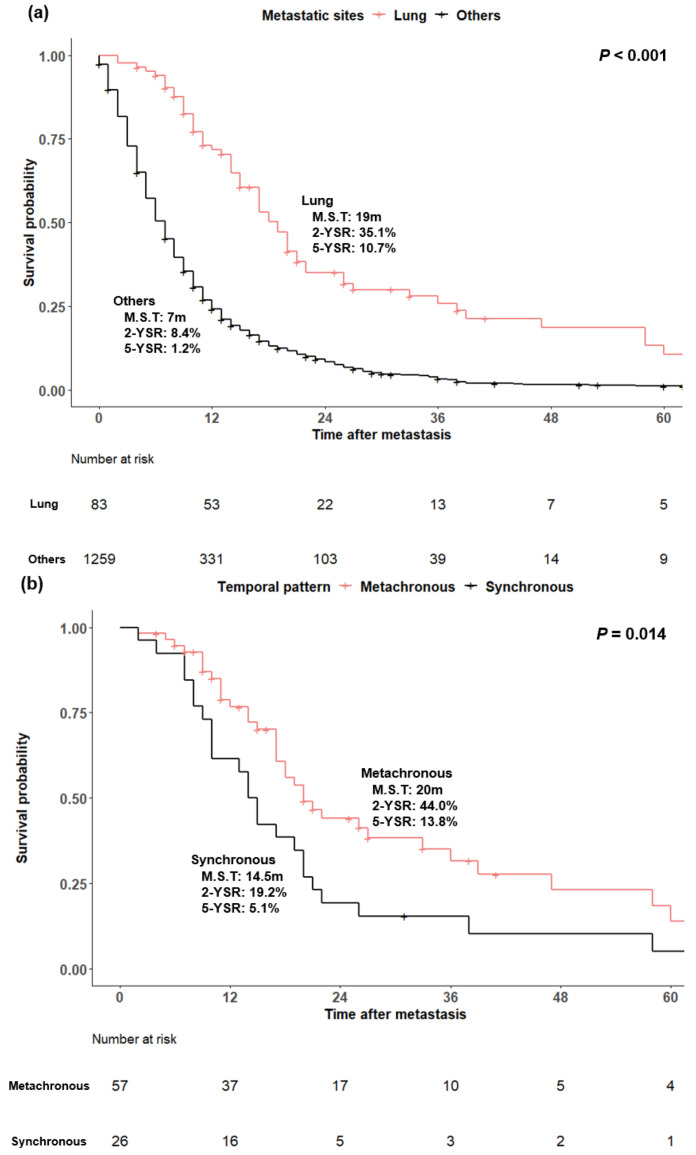
(**a**) Kaplan−Meier analysis of pancreatic ductal adenocarcinoma patients with distant metastasis according to metastatic sites. (**b**) Kaplan−Meier analysis of pancreatic ductal adenocarcinoma patients with isolated lung metastasis according to the temporal metastatic pattern. (**c**) Kaplan−Meier analysis of pancreatic ductal adenocarcinoma patients with isolated lung metastasis according to multiplicity of lung lesions. M.S.T., median survival time; YSR, year survival rate.

**Figure 4 cancers-14-02067-f004:**
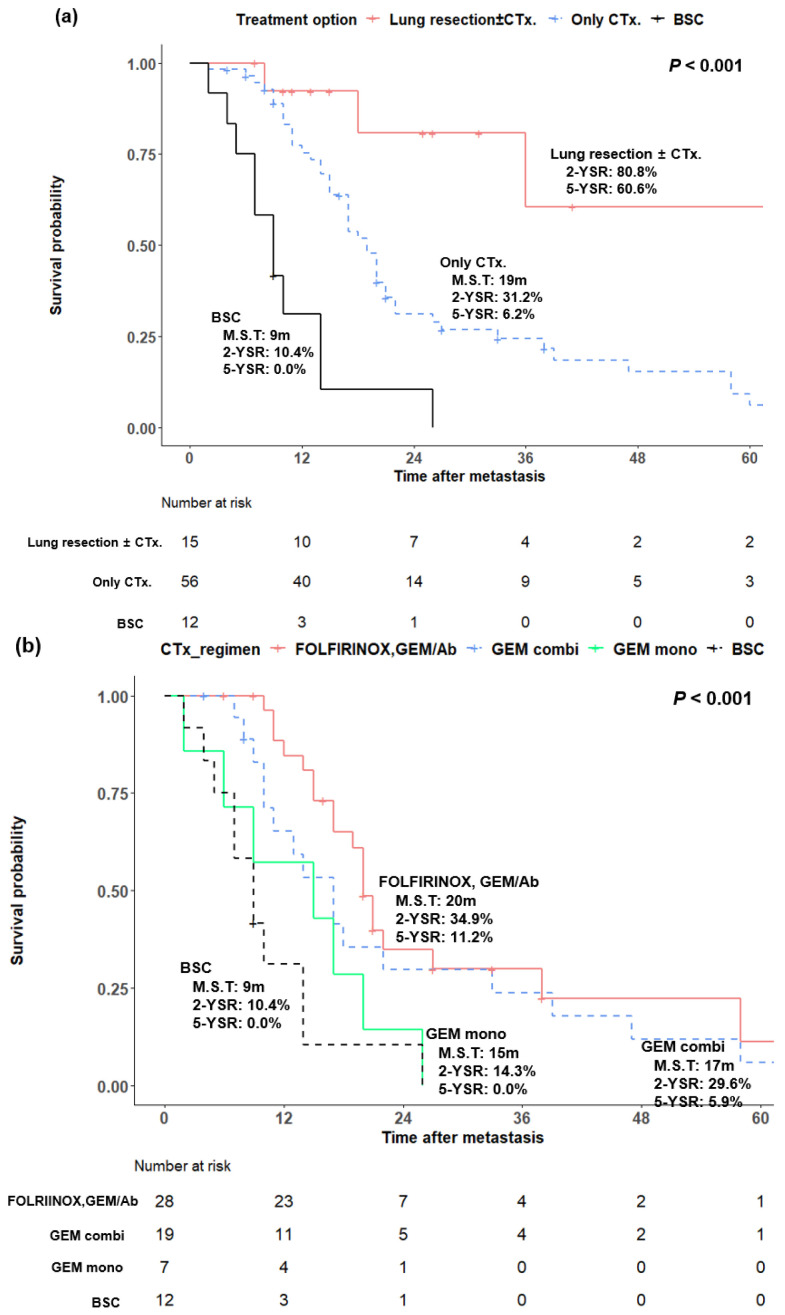
(**a**) Kaplan−Meier analysis of pancreatic ductal adenocarcinoma patients with isolated lung metastasis according to treatment options. (**b**) Kaplan−Meier analysis of pancreatic ductal adenocarcinoma patients with isolated lung metastasis who did not undergo metastatectomy according to a chemotherapy regimen. CTx., chemotherapy; BSC, best supportive care; M.S.T., median survival time; YSR, year survival rate; FOLFIRINOX, fluorouracil plus leucovorin, irinotecan, and oxaliplatin; GEM/Ab, gemcitabine plus albumin-bound paclitaxel; GEM combi, gemcitabine combined with agents other than albumin-bound paclitaxel; GEM mono, gemcitabine monotherapy.

**Figure 5 cancers-14-02067-f005:**
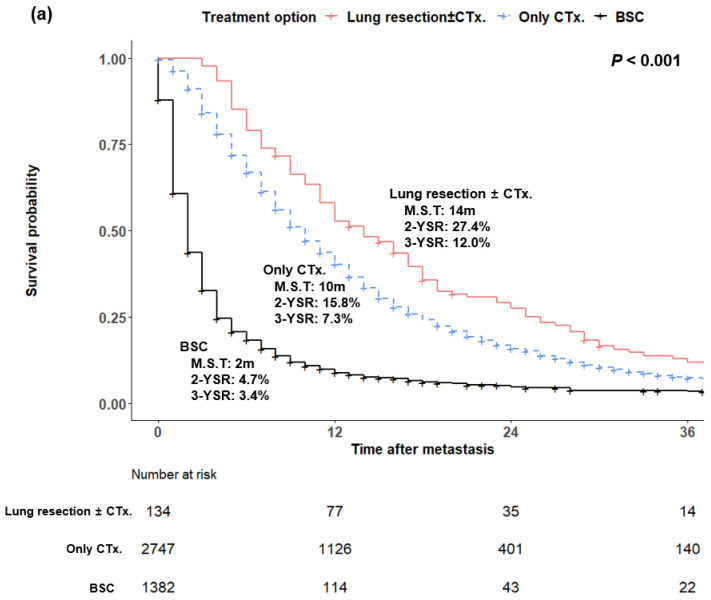
(**a**) Kaplan−Meier analysis of pancreatic ductal adenocarcinoma patients with isolated lung metastasis from the National Cancer Database according to treatment options. (**b**) Kaplan−Meier analysis of pancreatic ductal adenocarcinoma patients with isolated lung metastasis from the National Cancer Database who did not undergo surgery according to a chemotherapy regimen. CTx., chemotherapy; BSC, best supportive care; M.S.T, median survival time; YSR, year survival rates.

**Table 1 cancers-14-02067-t001:** Demographics of PDAC patients with isolated pulmonary metastasis.

Variables	No. of Patients (*n* = 83)
Age, mean (SD), years	64.5 (10.3)
Sex (male)	39 (47.0%)
Tumor location (head)	42 (50.6%)
Tumor size (Pancreas), mean (SD), mm	28.1 (11.6)
Clinical N stage (cN+)	16 (19.3%)
CEA (ng/mL) > 5	19 (22.9%)
CA 19-9 (U/mL) > 37	55 (66.3%)
Temporal metastatic pattern	
Synchronous	26 (31.3%)
Metachronous	57 (68.7%)
Multiplicity	
Single	23 (27.7%)
Oligometastases	29 (34.9%)
Multiple	31 (37.3%)
Metastatectomy (yes)	15 (18.1%)
Chemotherapy	
No chemotherapy	14 (16.9%)
Gemcitabine monotherapy	10 (12.0%)
Gemcitabine combination therapy (except GEM/Ab)	20 (24.1%)
Gemcitabine/Abraxane	13 (15.7%)
FOLFIRINOX	24 (28.9%)
Others	2 (2.4%)
Chemotherapy cycle, median (range)	6 (0–40)
Response to chemotherapy	
PR	11 (16.2%)
SD	39 (57.4%)
PD	13 (19.1%)
Limited to evaluate	5 (7.4%)
Radiotherapy at lung lesion	1 (1.2%)

CEA, carcinoembryonic antigen; CA, carbohydrate antigen; GEM/Ab, gemcitabine plus albumin-bound paclitaxel; FOLFIRINOX, fluorouracil plus leucovorin, irinotecan, and oxaliplatin; PR, partial response; SD, stable disease; PD, progressive disease.

**Table 2 cancers-14-02067-t002:** Univariate and multivariate analysis regarding the factors influencing survival after the occurrence of metastasis.

Variables		Univariate Analysis		Multivariate Analysis	
		Hazard Ratio	*p*-Value	Hazard Ratio	*p*-Value
Age (Years)	>65	1.068 (0.633~1.801)	0.806		
	≤65				
Sex	Male		0.457		
	Female	1.217 (0.726~2.039)			
Location	Head	1.345 (0.802~2.255)	0.261		
	Body/Tail				
Tumor size	>28 mm	1.434 (0.830~2.480)	0.197		
	≤28 mm				
Clinical N	cN0		0.190		
	cN+	1.515 (0.814~2.821)			
CEA (ng/mL)	>5	1.466 (0.822~2.613)	0.195		
	≤5				
CA 19-9 (U/mL)	>37	1.438 (0.796~2.598)	0.229		
	≤37				
Chemotherapy	Yes	0.265 (0.131~0.537)	<0.001	0.434 (0.210~0.896)	0.024
	No				
CTx. Regimen					
No chemothreapy		1.000 (Reference)			
GEM mono		0.555 (0.212~1.452)	0.230		
GEM combi		0.291 (0.129~0.659)	0.003		
FOLFIRINOX		0.210 (0.097~0.456)	<0.001		
or GEM/Ab					
CTx. Cycle	>6	0.287 (0.162~0.508)	<0.001	0.300 (0.159~0.564)	<0.001
	≤6				
RTx. at Lung	Yes	3.388 (0.454~25.265)	0.234		
	No				
Lung resection	Yes	0.183 (0.058~0.586)	0.004	0.278 (0.084~0.920)	0.036
	No				
Multiplicity	>5	1.811 (1.076~3.047)	0.025	2.004 (1.133~3.544)	0.017
(Lung lesion)	≤5				

CEA, carcinoembryonic antigen; CA, carbohydrate antigen; CTx., chemotherapy; GEM mono, gemcitabine monotherapy; GEM combi, gemcitabine combined with agents other than albumin-bound paclitaxel; FOLFIRINOX, fluorouracil plus leucovorin, irinotecan, and oxaliplatin; GEM/Ab, gemcitabine plus albumin-bound paclitaxel; RTx., radiotherapy.

## Data Availability

Not applicable.

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
