# Peer review of "Can Surgical Resection of Metastatic Lesions Be Beneficial to Pancreatic Ductal Adenocarcinoma Patients with Isolated Lung Metastasis?"

_cancers, 2022, doi:10.3390/cancers14092067_

Round 1

Reviewer 1 Report

Won-Gun et al provided a well-written and interesting study on the management of metastatic PDAC. Despite the retrospective nature of the study, the idea is original, it is focusing on an important issue, and definitely will be worth future confirmations in an RCT setting. 

Author Response

Response to Reviewer 1 Comments:

Reviewer 1.

  1. Won-Gun et al provided a well-written and interesting study on the management of metastatic PDAC. Despite the retrospective nature of the study, the idea is original, it is focusing on an important issue, and definitely will be worth future confirmations in an RCT setting.

Response: We would like to thank you for your positive evaluation of this article, and we also hope to a higher level of evidence from research in the RCT setting in the near future.

Reviewer 2 Report

Yun and colleagues reported “Can Surgical Resection of Metastatic Lesions be Beneficial to Pancreatic Ductal Adenocarcinoma Patients with Isolated Lung Metastasis?”. This study evaluated the effect of metastatectomy on survival after metastasis in PDAC patients with isolated lung metastasis. The authors concluded that metastatectomy with primary pancreatic lesions is recommended with effective chemotherapy. I have some comments about this manuscript.

  • The authors evaluated the effect of metastatectomy and chemotherapy on survival after metastasis in PDAC patients with the synchronous and metachronous lung metastasis. Because the prognosis of synchronous lung metastasis was significantly worse than that of metachronous lung metastasis in Figure 3b, the synchronous and metachronous lung metastasis seem to be analyzed separately.
  • In this study, lung resection was conducted before chemotherapy. How about perioperative safety and morbidity of the surgery? Several studies report a higher rate of pneumonitis in treatment of gemcitabine monotherapy and gemcitabine with other agents. Was chemotherapy administered safely?
  • As the authors mentioned, it is difficult to suggest an optimal timing of lung resection. How about recurrence free survival time and recurrence patterns after surgery?
  • It might be difficult to differentiate primary lung cancer from isolated lung metastasis. In such cases, VATS should be a good option. A single metastasis was observed in 23 cases, and oligometastases in 29 cases. Why was lung resection conducted in only 15 patients? Were other cases biopsied to confirm the diagnosis of lung metastases?

  • Page 11, line 43 (that tumor location within the pancreas ------------------the primary PDAC lesion). I don't understand what this sentence means.

Reviewer 3 Report

WG Yun et al. investigated the impact of lung metastasis on the prognosis of pancreatic adenocarcinoma based on two retrospective patient cohorts (1342 and 4263) that included respectively 83 and 134 cases of lung metastasis resections (associated or not with chemotherapy). Surgical resection of lung metastasis is significantly associated with a better prognosis (highly significant). In addition, the authors have identified negative (number of metastasis) and positive (lung metastasis resection, associated chemotherapy and number of cycles) prognosis factors.

The paper is clear and well written

Some concerns and questions:

  • Why this difference of prognosis between synchronous and metachronous lung metastasis?
  • Among molecular mechanisms and status it would be interesting to know the comparative status of Kras mutations between primary tumor and lung metastasis: that should be discussed in term of possible prognosis factor
  • The quality of survival curves (i.e. curves themselves) and some legends/notes inside the figure, otherwise confusing such as Fig 3C, 4B, 5B must be improved.

Round 2

Reviewer 2 Report

The revision is adequate.